Assessment of methods used for 3-dimensional superimposition of craniofacial skeletal structures: a systematic review

Mai Daniel Dinh-Phuc
Stucki Sven
Gkantidis Nikolaos nikosgant@yahoo.gr nikolaos.gkantidis@zmk.unibe.ch
Department of Orthodontics and Dentofacial Orthopedics, University of Bern , Bern , Switzerland
Koletsi Despina
Electronic publication date: 2020 Jun 5
Publication date: 2020
Volume: 8
Electronic Location ID: e9263
Received 2020 Mar 9; Accepted 2020 May 9
Copyright: ©2020 Mai et al.
Copyright year: 2020
Copyright holder: Mai et al.
License: This is an open access article distributed under the terms of the Creative Commons Attribution License, which permits unrestricted use, distribution, reproduction and adaptation in any medium and for any purpose provided that it is properly attributed. For attribution, the original author(s), title, publication source (PeerJ) and either DOI or URL of the article must be cited.
License URL: https://creativecommons.org/licenses/by/4.0/

Keywords: Cone-Beam Computed Tomography, Computed Tomography, Head, Superimposition, 3D imaging

Funding: The authors received no funding for this work.

==============================
Background

So far, several techniques have been recommended for the assessment of craniofacial changes through skeletal tissue superimposition, but the evidence that supports them remains unexplored. The purpose of the present study is to assess the available literature on skeletal-tissue superimpositions of serial craniofacial CT or CBCT images used to detect morphological changes.

Materials and Methods

Medline (via Pubmed), EMBASE, Google Scholar, Cochrane Library, Open Grey and Grey Literature Report were searched (last search: 17.11.2019) using specific terms that fulfilled the requirements of each database in the context of the study aim. Hand searches were also performed. The outcomes of interest were the accuracy, precision, or agreement between skeletal-tissue superimposition techniques to assess changes in the morphology of craniofacial structures. Studies of any design with sample size ≥3 were assessed by two authors independently. The study protocol was registered in PROSPERO (ID: CRD42019143356).

Results

Out of 832 studies, fifteen met the eligibility criteria. From the 15 included studies, 12 have shown high total risk of bias, one low risk of bias, and two studies have shown unclear risk of bias. Thirteen out of the 15 studies showed high applicability concerns, two unclear and no study had low applicability concerns. There was high heterogeneity among studies regarding the type of participants, sample size, growth status, machines, acquisition parameters, superimposition techniques, assessment techniques and outcomes measured. Fourteen of them were performed on Cone Beam Computed Tomography (CBCT) and one on Computed Tomography (CT) derived 3D models. Most of the studies (eleven) used voxel-based registration, one landmark-based registration and three studies compared different registration techniques, which include the surface-based registration. Concerning the area of interest, nine studies focused on the anterior cranial base and certain facial structures, four on maxillary structures and four on mandibular structures. Non-growing participants were included in six studies, growing in eight, whereas one study had both.

Conclusion

Most of the available studies had methodological shortcomings and high applicability concerns. At the moment, certain voxel-based and surface-based superimpositions seem to work properly and to be superior to landmark-based superimposition. However, further research in the field is required to develop and properly validate these techniques on different samples, through high quality studies with low applicability concerns.

Introduction

Superimpositions of serial craniofacial images have been widely applied in dental or other fields as a mean to depict changes over time. It is a valuable tool facilitating the better understanding of the effects of treatment or growth on dental and craniofacial morphology. The “structural method” developed by Björk (1969) and Björk & Skieller (1977) is still considered today as the standard 2D superimposition technique for craniofacial radiographs. This is based on reference structures that are considered stable, allowing the visual inspection of craniofacial changes relative to these stable structures. Such 2D methods were adopted and modified to be applicable in 3D data, since the same principles for reference area selection apply to both 2D and 3D approaches.

There are several advantages of the 3D over the 2D imaging techniques including the more accurate, real size information in all dimensions of space. Furthermore, the 3D information is not highly dependent on head positioning, which might be a critical source of error in 2D imaging. However, the higher amount and quality of 3D information does not come without a cost. This has to do mainly with the increased radiation dose needed to obtain the 3D images. Moreover, the data acquisition, handling, and processing of 3D data is usually more complex, time consuming, and expensive.

In growing individuals, the identification of stable superimposition reference areas might be more difficult, since the morphology of most craniofacial structures changes considerably over time. Thus, changes in the reference areas might affect the superimposition outcomes on the areas of interest, due to the incomplete matching of the first. For this reason, similar to the registration in 2D, the anterior cranial base still holds its position as the gold standard reference area, since its growth is more or less completed around the age of seven (Buschang et al., 1986; Afrand et al., 2014). To include this part of the head in the 3D images requires a larger field of view, and thus a larger dose of radiation. Furthermore, structures that are more vulnerable to radiation exposure, such as the eyes or certain brain structures, are included. Due to this fact, researchers have explored other possibilities, to substitute the anterior cranial base as a superimposition reference, which can be applied in smaller field of view scans and still perform properly (Nada et al., 2011; Gkantidis et al., 2015).

Following the development of such methods and due to the technological advancements, that can lead to 3D image acquisition with small radiation exposure, in the foreseeable future, large amount of reliable 3D data could be generated. This could facilitate the valid prediction of morphological changes that will occur in a specific patient after a certain treatment or growth occurrence, leading to individualised, less invasive and more efficient treatment strategies.

Since the first application of 3D superimposition, three main techniques have been used for serial image registration: namely landmark-based, surface-based, and voxel-based techniques (Grauer, Cevidanes & Proffit, 2009; Cevidanes et al., 2010; Almeida et al., 2011; AlHadidi et al., 2011). Each technique has been widely used for clinical and research purposes and has inherent limitations, advantages, and disadvantages. Furthermore, all techniques have been suggested in the literature to work properly.

Various relevant studies have been published so far, but the heterogeneity of the protocols, machines, acquisition parameters, and superimposition references did not allow for the development of solid conclusions (Ponce-Garcia et al., 2018). The only existing systematic evaluation of the literature included studies that were published prior to 2017 and regarded only the anterior cranial base (Ponce-Garcia et al., 2018). Thus, neither the accuracy, the precision, and the reproducibility of hard-tissue superimposition techniques nor the choice of reference structures have been thoroughly investigated recently. Hence, the purpose of this review is to provide a synopsis and a thorough assessment of the current evidence, aiming to provide guidelines for the proper use of the techniques and interpretation of the outcomes and identify fields where further research is needed.

Materials and Methods

Protocol and registration

The protocol was registered in PROSPERO prior to the study implementation (ID: CRD42019143356). This protocol consists a modification of a previously published protocol by Stucki & Gkantidis (2019) for an analogous, but fundamentally different topic.

Search strategy

The following databases were searched for eligible studies: Medline (via Pubmed), EMBASE, Google Scholar, Cochrane Library, OpenGrey and GreyLiteratureReport. The last search was performed on 17.11.2019, without time restriction. Unpublished literature was searched through the National Research Register, Pro-Quest Dissertation Abstracts and Thesis database, additional hand searches of all relevant studies were also performed. The specific search strategies applied for each database are provided as Appendix S1.

Selection criteria applied for the review

• Study design: Any study design, including prospective, and retrospective studies of any type.

• Study sample: Studies with sample size ≥ 3.

• Index test: 3D skeletal-tissue superimposition techniques to assess any change in the morphology of the craniofacial complex.

• Types of participants: Serial craniofacial CT or CBCT images of individuals or skulls who have received any kind of actual or simulated treatment, or whose craniofacial morphology is expected to be altered due to growth or pathology.

• Type of intervention: 3D skeletal-tissue superimposition to assess any morphological change in the craniofacial complex.

• Primary outcome: Superimposition accuracy or precision of a technique, or agreement between techniques measured in terms of angles or distances between specific skeletal or facial landmarks or area distances between corresponding models. Volume differences measured following 3D superimposition were also considered. Studies that evaluated any of the above parameters as a secondary outcome were also included.

• Comparator/control group: Studies that compared different superimposition techniques, direct measurements, or repeated measurements were selected.

• Unit of analysis: The measured distance, angle, or volume.

• Follow-up: Any observation period between subsequent models.

• Exclusion criteria: None.

Study selection

Following the search strategy, the selected databases were screened by two authors of the review (Daniel Dinh-Phuc Mai and Sven Stucki). There was no blinding concerning the authors’ names and affiliations, or the outcomes of the included studies. Titles and abstracts were evaluated first, if necessary the full text was read to evaluate the eligibility. The same authors read all eligible studies again in full text, independently, whereas non-eligible studies were excluded. Thereafter the eligibility was discussed between all team members until a consensus was reached, under the guidance of the last author (Nikolaos Gkantidis). A record of all decisions made during this process was retained.

Data extraction

The first and the last author performed data extraction independently and in duplicate, aiming to extract from the eligible studies the following information:

• Methods: Author, title, year, objectives, and design of study.

• Participants: Patient number, age, and gender.

• Materials: 3D model acquisition method and time between serial models.

• Superimposition method: Type of superimposition reference areas or points and software with specific settings used.

• Comparison/control group: Type and characteristics.

• Outcome: Type of outcome(s) and method of outcome assessment.

• If necessary, the authors were contacted by email to request missing data. If the relevant information was not provided, only the available information was used.

Assessment of heterogeneity

Study characteristics, similarity between types of participants, compared methods and assessed outcomes were considered to define heterogeneity among studies.

Assessment of reporting bias

We conducted an accurate, but also broad enough search of multiple sources, including on-going studies, to minimize potential reporting biases, such as publication bias and duplicate reports.

Data synthesis

A meta-analysis will be performed if there are at least two studies graded with an unclear or a low risk of bias and additionally use similar methods or report the same outcomes measured on similar data.

Subgroup analysis

Results will also be tested for the following factors, if possible:

• CBCT vs. CT data.

• Growing vs. non-growing patients.

• Short-term (within 1 year) vs. medium/long-term (> 1 year) interval between serial models.

• Superimposition on the anterior cranial base vs. superimposition on maxillary structures vs. superimposition on mandibular structures

Quality assessment

The quality of the selected studies was evaluated using the QUADAS-2 tool (Whiting, 2011). This is a widely used tool to evaluate the diagnostic accuracy of methods in systematic reviews. Using the QUADAS-2 tool the patient selection, the index test, the reference standard and the flow and timing are evaluated regarding their risk of bias and applicability concerns. Usually, gradings are shown in a table using happy (low risk) or sad smiles (high risk). In case an evaluation is not possible, e.g., because of missing data, an interrogation mark is shown (unclear risk). The total risk of bias or applicability concerns of each study correspond to the worst rating given in the individual items assessed each time.

Figure 1 Flowchart of study selection according to PRISMA guideline.

The quality assessment of all studies was performed by two authors (Daniel Dinh-Phuc Mai and Nikolaos Gkantidis) independently. If there was a disagreement, a consensus was reached through discussion among all authors. Studies graded with a high risk of bias were not to be included in a meta-analysis.

Results

Description of studies

The search results are shown in Fig. 1. After searching various databases, 2,540 studies were found. Seven additional studies were identified through hand searches. After removing the duplicates, 832 studies remained. These studies were screened by reading the titles and abstracts. Full-text reading of 24 studies was performed to evaluate the eligibility. Nine studies did not match the review question and thus, they were excluded as irrelevant to the study topic. Following the selection process 15 studies were included in this review.

All included studies used 3D skeletal-tissue superimposition techniques to assess morphological changes in the craniofacial complex, the accuracy or precision of the applied processes, or the agreement between different techniques as a primary outcome.

Quality assessment

The quality assessment of the included studies is provided in Table 1.

From the 15 included studies, 12 of these have shown a high total risk of bias, one a low risk of bias, and 2 studies have shown an unclear risk of bias. Regarding the individual items 4 studies have high, 7 low, and 4 unclear risk of bias in the patient selection. Regarding the index test, 8 studies have high, 6 low, and one unclear risk of bias. The reference standard of 9 studies shows a high risk of bias, of 2 low, and of 4 unclear. The flow and timing of 2 studies has high, of 11 low, and of 2 unclear risk of bias.

Thirteen out of the 15 studies showed high total applicability concerns, 2 unclear and no study had low applicability concerns. Concerning the individual items, 6 studies had high, 6 unclear and 3 low applicability concerns in the patient selection. Regarding the index test, 9 studies had high, one unclear and 5 low applicability concerns. The reference standard of 9 studies showed high, of 3 unclear and of 3 low applicability concerns.

Table 1 Quality assessment of the included studies through the QUADAS-2 tool.

	Risk of Bias	Applicability Concerns	
Study	Patient selection	Index test	Reference standards	Flow & timing	Total risk of bias	Patient selection	Index test	Reference standards	Total applicability concerns	
Almukhtar et al.,
PLoS One (2014)	☺	☺	☺	☺	☹	☺	☹	☹	☹	
Bazina et al.,
Am J Orthod Dentofacial Orthop (2018)	☺	☹	☹	☺	☹	☺	☹	☹	☹	
Cevidanes et al.,
Dentomaxillofac Radiol (2005)	?	☺	☺	☺	?	?	☺	☺	?	
Cevidanes et al.,
Am J Orthod Dentofacial Orthop (2009)	☹	☺	?	☺	☹	☹	☺	?	☹	
Gkantidis et al.,
PLoS One (2015	☺	☺	☺	☺	☺	☹	☺	☺	☹	
Ghoneima et al.,
Orthod Craniofac Res (2017)	☹	☹	☹	☹	☹	☹	☹	☹	☹	
Häner et al.,
Orthod Craniofac Res (2019)	?	?	?	☺	?	?	?	?	?	
Koerich et al.,
Int J Oral Maxillofac Surg (2016)	☺	☺	☹	☺	☹	☹	☺	☹	☹	
Koerich et al.,
Angle Orthod (2017)	?	☺	☹	?	☹	☹	☺	☺	☺	
Lemieux et al.,
Am J Orthod Dentofacial Orthop (2014)	?	☹	☹	?	☹	?	☹	☹	☹	
Nada et al.,
PLoS One (2011)	☺	☹	?	☺	☹	?	☹	☺	☹	
Nguyen et al.,
Am J Orthod Dentofacial Orthop (2017)	☹	☹	☹	☺	☹	☹	☹	☹	☹	
Ruellas et al.,
Am J Orthod Dentofacial Orthop (2016a)	☺	☹	☹	☹	☹	☺	☹	☹	☹	
Ruellas et al.,
PLoS One (2016b)	☺	☹	☹	☹	☹	?	☹	☹	☹	
Weissheimer et al.,
Int J Oral Maxillofac Surg (2015)	☹	☹	?	☺	☹	☹	☹	?	☹	
Notes.

☺ low risk of bias/low applicability concerns.

☹ high risk of bias/high applicability concerns.

? unclear risk of bias/unclear applicability concerns.

Characteristics of the included studies

One of the included studies utilized prospective radiographic image acquisition and 14 a retrospective one. Regarding the superimposition data generation and method comparison all studies were prospective. Eight studies included only growing patients, 6 only non-growing and 1 study both. None of the eligible studies was performed in patients with severe craniofacial malformations, such as those related to systemic conditions, congenital anomalies, or syndromes. Fourteen studies were performed on Cone Beam Computed Tomography (CBCT) and 1 on Computed Tomography (CT) images. Eleven studies used voxel-based registration, 1 landmark-based registration, and 3 compared different registration techniques. Concerning the area of interest, 9 studies focused on the anterior cranial base and certain facial structures, 4 on maxillary structures and 4 on mandibular structures.

The characteristics of the included studies are provided in detail in Tables 2, 3 and 4.

Results and Qualitative synthesis of the included studies

The results of the included studies are shown in Table 4 and the conclusions and limitations in Table 5.

There was high heterogeneity among studies regarding the type of participants, sample size, growth status, machines, acquisition parameters, superimposition techniques, assessment techniques and outcomes measured. Therefore, no quantitative synthesis was performed.

For the qualitative synthesis, the included studies are categorized in three groups based on the registration technique assessed: 1. voxel-based registration, 2. landmark-registration and 3. comparison of different registration techniques, which include the surface-based registration.

Voxel-based registration

Eleven studies tested the voxel-based registration. Six of those studies included only growing patients, 4 only non-growing and one study included both. Nine studies of this subgroup had high, and 2 unclear risk of bias. Similarly, 9 studies had high and 2 unclear applicability concerns. Six studies used cranial base structures as superimposition reference, whereas 2 studies used maxillary and 4 mandibular sites.

Bazina et al. (2018) superimposed CBCTs of 31 non-growing patients on the anterior cranial base to evaluate the reproducibility of Dolphin voxel-based superimposition and its agreement with ITK-Snap+3D Slicer superimposition. The Dolphin 3D software seemed to work properly, but the study showed important limitations, high risk of bias, and high applicability concerns.

Cevidanes et al. (2005) tested the reproducibility of 3D cranial base superimpositions for the evaluation of mandibular ramus changes in maxillary orthognathic surgery patients. To verify reproducibility, changes from pre- to post-treatment were measured on mandibular areas of 10 non-growing patients. The surgery was performed exclusively on the maxilla and the assessments on the mandible. Hence, no or minimal changes are expected in the mandible. Under these circumstances, the technique showed acceptable reproducibility, though in certain cases the inter-observer variation was relatively high, compared to the limited original changes. The study had unclear risk of bias and applicability concerns.

Table 2 Main general characteristics of the included studies.

Study	Study objectives	Study design	Type of participants	Sample size	Growth status	Time span	
Almukhtar et al.,
PLoS One (2014)	To compare the trueness of voxel-based registration and surface-based registration for 3D assessment of surgical change following orthognathic surgery.	Retrospective (radiographs) // prospective methodological study	pre- & post-orthognatic surgery CBCTs	31 Patients	Non-growing	min. 6 months	
Bazina et al.,
Am J Orthod Dentofacial Orthop (2018)	To evaluate the reproducibility of Dolphin voxel-based superimposition and its agreement with ITK-Snap+3D Slicer superimposition.	Retrospective (Scans) // prospective methodological study	Pre- and post-1-jaw or 2-jaw orthognatic surgery including LeFort I osteotomy, bilateral sagittal split osteotomy, or genioplasty CBCTs	31 Patients	Non-growing (21 ± 8 years, range: 15-47 years)	13 months (within 1 month prior surgery and 12 months after surgery)	
Cevidanes et al.,
Dentomaxillofac Radiol (2005)	To determine the reproducibility of voxel-based superimposition to evaluate mandibular ramus changes in maxillary orthognatic surgery patients.	Prospective methodological study	Pre- and post-orthognatic surgery CBCTs	10 Patients	Non-growing	1 week	
Cevidanes et al.,
Am J Orthod Dentofacial Orthop (2009)	To determine the reproducibility of voxel-based superimpositions to evaluate overall facial changes in growing patients.	Retrospective (radiographs) // prospective methodological study	Pre- and post-orthopedic treatment of Class III malocclusion with miniplates CBCTs	3 Patients	Growing (mean age: 11.4 years)	1 year	
Gkantidis et al.,
PLoS One (2015)
	To test the applicability, trueness, precision, and reproducibility of various 3D superimposition techniques for radiographic data, transformed to triangulated surface data.	Retrospective (radiographs) // prospective methodological study	Pre- and post-rapid maxillary expansion CTs	8 Patients	Non-growing (median age: 16.2 years)	10–23 days	
Ghoneima et al.,
Orthod Craniofac Res (2017)	To evaluate the reproducibility of landmark-based, surface-based and voxel-based superimpositions, as well as their performance in matching duplicated scans.	Retrospective (CBCT images) // prospective methodological study	Pre- and post-correction of Class II malocclusion with Herbst appliance CBCTs	20 Patients (9 males, 11 females)	Growing (range: 8-15 years)	NA	
Häner et al.,
Orthod Craniofac Res (2019)	To evaluate the trueness, reproducibility and segmentation effect on hard tissue outcomes using voxel-based superimposition.	Retrospective (CBCT images) // prospective methodological study	Orthodontic patients without accounting for performed treatment or skeletal growth pattern CBCTs	15 Patients (8 males, 7 females)	Growing (11.75 ± 0.59 years)	1.69 ± 0.37 years	
Koerich et al.,
Int J Oral Maxillofac Surg (2016)	To evaluate the reproducibility of a superimposition method for the maxilla and mandible in non-growing patients.	Retrospective (radiographs) // prospective methodological study	1. Two serial CBCT images of dry skulls after changing their position
2. Two serial CBCT images of orthodontic or wisdom tooth surgery patients	1. 2 Dry skulls
2. 15 Patients	Non-growing	12.3 months (range: 4–24 months)	
Koerich et al.,
Angle Orthod (2017)	To evaluate the reproducibility of a voxel-based superimposition of the mandible in growing patients.	Retrospective (scans) // prospective methodological study	Pre- and post-rapid palatal expansion CBCTs	24 Patients	Growing (mean age: 10.8 ± 1.7 years)	16 ± 2.9 months	
Lemieux et al.,
Am J Orthod Dentofacial Orthop (2014)	To evaluate the trueness of a maxillary superimposition plane using the nasomaxillary complex as reference.	Retrospective (CBCT images) // prospective methodological study	Pre- and post-rapid palatal expansion CBCTs	30 Patients	Growing (dental age of 12)	within 12 months	
Nada et al.,
PLoS One (2011)	To evaluate the trueness and reproducibility of a semi-automated voxel-based registration on two regions: 1. anterior cranial base and 2. zygomatic arches	Retrospective (radiographs) // prospective methodological study	Pre- and-post-orthognatic surgery CBCTs	16 Patients	Non-growing (mean age: 26 ± 9 years)	18 ± 4.6 months	
Nguyen et al.,
Am J Orthod Dentofacial Orthop (2017)	1. To identify stable anatomical regions in the mandible.
2. To evaluate the reproducibility of the chin+symphysis registration.	Retrospective (CBCT images) // prospective methodological study	1. CBCTs of 20 Class III patients with bone plates and screws in the mandibular anterior area
2. Pre- and post-correction of Class II with Herbst appliances CBCTs (n = 10); Pre- and post-correction of Class II with elastics CBCTs (n = 10); Pre- and post-correction of Class III with bone anchors CBCTs (n = 5)	25 Patients
	Growing (mean age: 12.7 ± 1.4 years)	1. 1.2 years
2.12.6 ± 0.9 months	
Ruellas et al.,
Am J Orthod Dentofacial Orthop (2016a)	To evaluate the differences between voxel-based registration on 2 regions of the maxilla (1. Maxillary region and 2. Palate and Infrazygomatic region) and the reproducibility of each technique	Retrospective (radiographs) // prospective methodological study	Pre- and post-rapid maxillary expansion for crossbite correction (n = 8) and Pre- and post-correction of Class II malocclusion with Herbst appliance (n = 8)	16 Patients	Growing (9–13 years)	6 months	
Ruellas et al.,
PLoS One (2016b)	To evaluate superimposition of serial mandibular models on 3 reference regions (Björk, modified Björk and mandibular body) as compared to directly measured changes in interlandmark distances.	Retrospective (radiographs) // prospective methodological study	NA	16 Patients	growing (9–13 years)	min. 18 months	
Weissheimer et al.,
Int J Oral Maxillofac Surg (2015)	To evaluate the trueness of a voxel-based superimposition technique using the anterior cranial base as reference for growing and non-growing patients	Retrospective (radiographs) // prospective methodological study	1. Pre-treated images reoriented and superimposed on the original (n = 10)
2. Pre- and post-orthognatic surgery (n = 4)
3. Pre- and post-rapid palatal expansion (n = 4)
Time span: 1 year	18 Patients	1. Growing (11.4 ± 1 year)
2. Non-growing (26.3 ± 5.7 years)
3. Growing (9.5 ± 1.8 years)	1 year	

Table 3 Main superimposition-related characteristics of the included studies.

Study	Superimposition methods	References	No of Operators	Machines	Acquisition parameters	Software	
Almukhtar et al., PLoS One (2014)	Voxel-based registration (iterative best match of grey scale intensities)
Surface-based registration (iterative closest point)	VBR: Anterior cranial base (extended to involve the frontal bone) and forehead region (including the forehead and the eyes)
SBR: Anterior cranial base (for the hard tissue) and forehead region (for the soft tissue)	NA	CBCT: i-CAT Classic (Imaging Sciences, Hatfield, UK)	NA	Maxilim software (Medicim-Medical Image Computing, Belgium) for voxel-based registration (VBR).
VRMesh software (VirtualGrid, Bellevue City, WA) for surface-based registration (SBR).	
Bazina et al.,
Am J Orthod Dentofacial Orthop (2018)	Voxel-based Registration (approximation using 3 landmarks located at the right and left frontozygomatic sutures and the left mental foramen)	Voxel-based Registration (iterative best match of grey scale intensities)	1	CBCT: CB MercuRay scanner (Hitachi Medical Systems America Inc, Twinsburg, OH)	Tube voltage: 120 kVp; Tube current: 15 mA; FOV: 12-in; Grey scale 4096; Voxel size: 0.38 mm3; Exposure: 9.5 s	1. Dolphin 3D software (version 11.8.06.15 premium; Dolphin Imaging, Chatsworth, Calif) for the registration of T2 CBCT image to T1.
2. ITK-SNAP software program (version 3.0.0; http://www.itksnap.org) and 3D Slicer (version 4.4.0; http://www.slicer.org) for DICOM files conversion, segmentation of the area of cranial base and image registration.	
Cevidanes et al.,
Dentomaxillofac Radiol (2005)	Voxel-based Registration (iterative best match of grey scale intensities)	Cranial base	3
	CBCT: NewTom 9000 (Aperio Services LLC, Sarasota, FL, 34236)	FOV: 23x23 cm; Exposure: 70 s	MIRIT Software for the fully automated rigid registration.
VALMET Software for the 3D models comparison.	
Cevidanes et al.,
Am J Orthod Dentofacial Orthop (2009)	Voxel-based Registration (iterative best match of grey scale intensities)	Anterior cranial base	3
	CBCT: iCat (Imaging Sciences International, Hatfield, PA)	FOV: 16x22 cm; Voxel size: 0.5 mm3; Exposure: 40 s	Imagine software (http://ia.unc.edu/dev/download/imagine/index.htm) for the rigid registration.	
Gkantidis et al.,
PLoS One (2015	Surface-based registration (iterative closest point)	1. Three point registration (3P); 2. One zygomatic arch (1Z); 3. Both zygomatic arches (BZ); 4. Anterior cranial base (AC: body and small wing of the sphenoid bone and part of the bottom of the anterior cranial fossa); 5. Anterior cranial base + Foramen magnum (middle posterior part of the edge of the foramen magnum) (AC+F)	3	CT: Philipps Brillance 16 CT Scanner	Tube voltage: 120 kV; Tube current: 293 mA; FOV: 21x21x12 cm; Voxel size: 0.3mm3; Exposure: 2.5 s; Slice thickness: 0.8 mm; Spacing between slices: 0.4 mm; Spatial resolution: 16 lp/cm	Geomagic Qualify 2012 software for Windows (Geomagic GmbH, Stuttgart, Germany) for data conversion, model processing, registration, and 3D analysis.	
Ghoneima et al.,
Orthod Craniofac Res (2017)	1. Landmark-based Registration
2. Surface-based Registration (iterative closest point)
3. Voxel-based Registration (iterative best match of grey scale intensities)	1. Seven homologous points on the frontal and zygomatic bones
2. Anterior cranial base surface
3. Anterior cranial base (anterior wall of frontal sinus anteriorly, the anterior clinoid process posteriorly, the superior wall of ethmoid sinus superiorly and the inferior floor of sphenoid sinus inferiorly)	NA	CBCT: iCAT 3D imaging System (Imaging Sciences International, Hatfield, PA, USA)	Tube voltage: 120 kV; Tube current: 20 mA; FOV: 17 × 23 cm; Voxel size: 0.3 mm3; Exposure: 8.9 s	1 and 3: Dolphin software version 11.8 Premium (Dolphin Imaging and Management Solutions, Chatsworth, CA, USA) for the registration.
2: 3dMD Vultus software (3dMD, Atlanta, GA, USA) for the registration.	
Häner et al.,
Orthod Craniofac Res (2019)	Voxel-based Registration (iterative best match of grey scale intensities)	Anterior cranial base (from the middle of the sella turcica to the posterior wall of the sinus frontalis. The vertical height of the area is about 3.5 cm. The lower vertical limit was set 2-4 mm below the lowest point of the sella turcica. The lateral limits extend till the lateral walls of the cranium)	2	CBCT: KaVo3D eXam (Hatfield, PA 19440, USA)	Tube voltage: 120 kV; Tube current: 5 mA; FOV: 170 height mm x 232 mm; Voxel size: 0.4 mm3; Scan time: 8.9 s; Exposure: 3.7 s	Dolphin 3D software (version 2.1.6079.17633) for surface model creation and the voxel-based registration.
Viewbox 4 software (version 4.1.0.1 BETA 64) for surface model processing and analysis.	
Koerich et al.,
Int J Oral Maxillofac Surg (2016)	Voxel-based Registration (iterative best match of grey scale intensities)	Maxilla (zygomatic process and palate) and Mandible (Symphysis, corpus and part of ramus)	2
	1. CBCT: Kodak Carestream 9300 (Carestream Health Inc., Rochester, NY, USA)
2. CBCT: i-CAT scanner (Imaging Sciences International LLC, Hatfield, PA, USA)	1. Tube voltage: 85 kVp; Tube current: 4 mA; FOV: 13.5x17 cm; Voxel size: 0.3mm3; Exposure: 11.3 s
2. Tube voltage: 120 kVp; Tube current: 8 mA; FOV: 16x13 cm; Voxel size: 0.25 mm3; Exposure: 27 s	OnDemand 3D software v1.0.10.5261 (Cybermed, Seoul, Korea) for image processing, segmentation and registration.
VAM software (Canfield Scientific, Fairfield, NJ, USA) for analysis.	
Koerich et al.,
Angle Orthod (2017)	Voxel-based Registration (iterative best match of grey scale intensities)	Lower mandibular border below to tooth apices, extending from the middle of the symphysis to the distal of the first molars	2	CBCT: i-CAT scanner (Imaging Sciences International, Hatfield, PA)	Tube voltage: 120 kVp; Tube current: 8 mA; Voxel size: 0.3 mm3; Exposure: 40 s	OnDemand 3D software v1.0.10.5261 (Cybermed, Seoul, Korea) for image processing, segmentation and registration.
VAM software (Canfield Scientific, Fairfield, NJ, USA) for analysis.	
Lemieux et al.,
Am J Orthod Dentofacial Orthop (2014)	Landmark-derived plane Registration	Maxillary superimposition plane formed by nasion, bilateral infraorbital foramina and incisive foramen	1	CBCT: NewTom 3G volumetric scanner (Aperio, Verona, Italy)	Tube voltage: 110 kV; Tube current: 6.19 mAs; Voxel size: 0.25 mm3; Thickness Aluminium filtre: 8 mm	MATLAB software (R2008a; MathWorks, Natick, Mass) for landmarks-based registration.
Avizo software (version 6.0; Visualization Sciences Group, Burlington, Mass) for landmark location and analysis.	
Nada et al.,
PLoS One (2011)	Voxel-based Registration (iterative best match of the grey scale intensities)	1. Anterior cranial base (AC)
2. Left zygomatic arch (ZL)	2
	CBCT: i-CAT 3D Imaging System (Imaging Sciences International INC, Hatfield, PA, USA)	FOV: 22x16 cm; Voxel size: 0.4 mm3	Maxilim software (Medicim, Mechelen, Belgium) for 3D model construction, superimposition and analysis	
Nguyen et al.,
Am J Orthod Dentofacial Orthop (2017)	Voxel-based Registration (iterative best match of grey scale intensities)	1. Bony plates and mini-screws in the mandibular anterior area
2. Chin (anterior surface of the chin bounded vertically from pogonion to B-point and laterally at the distal-incisal point of the right and left lateral incisors) + Symphysis (internal cortical bone of the mandibular symphysis at the lateral limit of its lingual surface and from its inferior border to the level of the center of both mental foramina)	2
	CBCT: i-CAT machine (Imaging Sciences International, Hatfield, PA)
CBCT: NewTom 3G (AFP Imaging, Elmsford, NY)	Tube voltage: 12 kV(p); Tube current: 5 mA; Voxel size: 0.3 mm3; Exposure: 20–25 s	ITK-SNAP software (version 3.6; open-source software, http://www.itksnap.org) for 3D mandibular models creation.
Slicer CMF software (version 3.1; http://www.slicer.org) to create surface models and registration.	
Ruellas et al.,
Am J Orthod Dentofacial Orthop (2016a)	Voxel-based Registration (iterative best match of the grey scale intensities)	1. Maxillary region (maxillary bone clipped inferiorly at the dentoalveolar processes, superiorly at the plane passing through the right and left orbitale points, laterally at the zygomatic processes through the orbitale point, and posteriorly till the distal surface of the second molars)
2. Palate and Infrazygomatic region (same as above cropped posteriorly distal aspects to the first molars and anteriorly at the canines)	2	CBCT: i-Cat machine (Imaging Sciences International, Hatfield, PA)	FOV: 16 × 22 cm; Voxel size: 0.4 mm3	Slicer software (version 4.3.1; http://www.slicer.org) for the registration
ITK-SNAP software (http://www.itksnap.org) for data conversion and 3D models construction.
VECTRA Anaylsis Module software (version 3.7.6; Canfield Scientific, Fairfield, NJ) for landmark generation and analysis.	
Ruellas et al.,
PLoS One (2016b)	Voxel-based Registration (iterative best match of grey scale intensities)	1. Maxillary region (maxillary bone clipped inferiorly at the dentoalveolar processes, superiorly at the plane passing through the right and left orbitale points, laterally at the zygomatic processes through the orbitale point, and posteriorly at a plane passing through the distal surface of the second molars)
2. Palate & infrazygomatic region (same as above cropped posteriorly at the plane passing through the distal aspects of the first molars and anteriorly at the canines)	NA	CBCT: NA	FOV: 16x22 cm; Voxel size: 0.4 mm3	Slicer software (v4.4; http://www.slicer.org) for data analysis and registration.
ITK-SNAP software (http://www.itkspnap.org) for the segmentation.	
Weissheimer et al.,
Int J Oral Maxillofac Surg (2015)	Voxel-based Registration (iterative best match of grey scale intensities)	Anterior cranial base	NA
	CBCT: iCat (Imaging Sciences International, Hatfield, PA)	Tube voltage: 120 kVp; 8 mA; FOV: large; Voxel size: 0.25 mm3; Exposure: 40 s	OnDemand 3D software v1.0.10.5261 (Cybermed, Seoul, Korea) for the registration.	

In another study, Cevidanes et al. (2009) performed 3D superimpositions on the anterior cranial base to investigate the reproducibility of the technique for the evaluation of overall facial changes in three growing patients. Nine regions distributed on the whole face were assessed by three operators. Detailed results acquired by each operator were not reported and only the ranges were provided. Within this limitation, this method seemed reproducible in growing patients. However, as the sample size was quite small and did not allow statistical comparisons, no rigid conclusion can be made. This study showed high risk of bias and applicability concerns.

Häner et al. (2020) evaluated the trueness, reproducibility, and segmentation effect on hard tissue outcomes using the Dolphin voxel-based superimposition. Fifteen growing patients were included, and the superimposition was performed on the anterior cranial base. The trueness of the voxel-based superimposition was assessed through visual inspection of corresponding reference structures, and the intra and inter-operator reproducibility was assessed through repeatedly superimposed 3D models. The superimposition technique exhibited adequate performance in growing patients, in terms of efficiency, cranial base matching, and reproducibility. The segmentation error was also acceptable in most cases. However, due to certain limitations the study showed unclear risk of bias and applicability concerns.

Table 4 Results of the included studies.

Study	Main Outcomes	Secondary Outcomes	Main Results	Secondary Results	
Almukhtar et al.,
PLoS One (2014)	Mean absolute distance of surface models in unchanged areas (anterior cranial base for hard tissue and forehead for soft tissue models): 1. VBR hard; 2. VBR soft; 3. SBR hard; 4. SBR soft	Correlation between VBR and SBR results on hard and soft tissues	Mean absolute distances (mm): 1. 0.050 ± 0.206; 2. 0.294 ± 0.334; 3. 0.047 ± 0.259; 4. 0.230 ± 0.561
VBR hard - SBR hard (p = 0.392)
VBR soft - SBR soft (p = 0.243)	VBR hard - SBR hard: r = 0.886
VBR soft - SBR soft: r = 0.126	
Bazina et al.,
Am J Orthod Dentofacial Orthop (2018)	1. Reproducibility of the Dolphin technique
2. Agreement with the ITK-Snap+3D Slicer assessed through the mean differences at 7 areas: a. Nasion area; b. A-point area; c. Right zygomatic area; d. Left zygomatic area; e. Right gonial angle; f. B-point area; g. Left gonial angle	NA	1. ICC = 0.964 (0.941 - 0.978)
2. Mean differences (mm) = a. 0.099 ± 0.072; b. 0.188 ± 0.110; c. 0.113 ± 0.086; d. 0.092 ± 0.057; e. 0.210 ± 0.136; f. 0.189 ± 0.101; g. 0.169 ± 0.082	NA	
Cevidanes et al.,
Dentomaxillofaci Radiol (2005)	Inter-operator agreement on surface distance measurements of 3D models at 3 mandibular regions: 1. Anterior mandibular ramus, 2. Posterior mandibular ramus, 3. Condyles	NA	Surface distances (mm): 1. 0.25 ± 0.11; 2. 0.13 ± 0.05; 3. 0.09 ± 0.05	NA	
Cevidanes et al.,
Am J Orthod Dentofacial Orthop (2009)	Inter-operator agreement on surface distance measurements of 3D models at 9 regions: 1. Zygomatic process, 2. Anterior maxilla, 3. Chin, 4. Right anterior condyle, 5. Right posterior condyle, 6. Left anterior condyle, 7. Left posterior condyle, 8. Mandibular inferior border, 9 Soft-tissue upper lip	NA	Surface distances (mm): 1. 0.1–0.4; 2. 0.2 - 0.5; 3. 0.1 - 0.4; 4. 0.0 - 0.3; 5. 0.1–0.4; 6. 0.0–0.3; 7. 0.0–0.4; 8. 0.2 - 0.4; 9. 0.3 - 0.5	NA	
Gkantidis et al.,
PLoS One (2015)	A. Trueness (overall deviation of surface models at unchanged areas: AC + F)
B. Intra-operator agreement (on measured structural changes at four corresponding landmarks) of different superimposition techniques: 1. 3P; 2. 1Z; 3. BZ; 4. AC; 5. AC+F
C. Inter-operator agreement assessed as described above	NA	A. Trueness (median values of the 3 operators in mm): 1. 0.79 - 1.01; 2. 1.42 - 1.76; 3. 0.31 - 0.57; 4. 0.35 - 0.52; 5. 0.07 - 0.11 (p = 0.0002)
B. p = 0.854
C. p = 0.661; r > 0.91 for all except 3P	NA	
Ghoneima et al.,
Orthod Craniofac Res (2017)	A. Reproducibility of each superimposition technique
B. Mean absolute distance between manually located landmarks on superimposed duplicated scans (ACP, Ba-x, Ba-y, PNS-y, B point-x, Me-x, U1-x, L1-x)	NA
	Surface-based and Voxel-based superimposition methods using the anterior cranial base as reference seem to be reproducible whereas Landmarks-based superimposition is less reproducible.	NA	
Häner et al.,
Orthod Craniofac Res (2019)	1. Trueness of the voxel-based superimposition assessed through visual inspection of corresponding reference structures
2. Intra-operator reproducibility assessed through the mean absolute distance (MAD) of the repeatedly superimposed T0 surface models measured in the following areas: N-point, A-point, Pogonion, Right and Left zygomatic arch, Right and left gonial angle
3. Inter-operator reproducibility assessed as described above	Segmentation effect (manual and automatic) assessed as the intra- and interoperator reproducibility	1. In all cases, visual inspection of the superimposed T0-T1 volumes presented adequate overlap
2. MAD (0.06 - 0.16 mm). In very few cases, it exceeded 0.5 mm and never 1 mm
3. MAD (0.15 - 0.24 mm). In few cases, it exceeded 0.5 mm and never 1.5 mm	The median segmentation error ranged from 0.05 - 0.12 mm. The biggest segmentation error was found at A-point (0.3 mm)	
Koerich et al.,
Int J Oral Maxillofac Surg (2016)	A. Intra-operator agreement on surface distance measurements (RMSD) of serial 3D models at 2 regions of the maxilla and 3 regions of the mandible (average difference)
B. Inter-operator agreement assessed as described above	NA
	A.1 Intra-operator agreement (mm): NA
A.2 Intra-operator agreement (mm). Maxilla: 0.183 - 0.184, Mandible: −0.005 - 0.001
B.1 Inter-operator agreement (mm). Maxilla: 0.087 - 0.098, Mandible: 0.183 - 0.184
B.2 Inter-operator agreement (mm). Maxilla: 0.072 - 0.092, Mandible: 0.087 - 0.105	NA	
Koerich et al.,
Angle Orthod (2017)	Inter-operator agreement on surface distance measurements (RMSD) at 5 mandibular regions: 1. Right mandible, 2. Chin, 3. Left mandible, 4. Right ramus, and 5. Left ramus, located at the outer surface of the mandible	NA	Surface distances (mm): 1. 0.11 ± 0.12; 2. 0.14 ± 0.1; 3. 0.11 ± 0.16; 4. 0.33 ± 0.29; 5. 0.36 ± 0.33		
Lemieux et al.,
Am J Orthod Dentofacial Orthop (2014)	Amount of expansion at the levels of the first premolars (from tip to tip of each buccal cusp) and the first molars (from tip to tip of each mesiobuccal cusp) on 1. plaster models and 2. 3D plane superimposition	Landmark identification reproducibility through ICC	Mean distances measured between premolars (mm): 1. 2.97 ± 2.12; 2. 3.06 ± 1.97
Mean distances measured between molars (mm): 1. 4.18 ± 1.62; 2. 4.28 ± 1.61	ICC > 0.924, 0.992, 0.973 in the x, y, and z axes respectively	
Nada et al.,
PLoS One (2011)	Mean absolute distance of surface models on the following stable areas: a. anterior cranial base (CB); b. forehead (FH); c. left zygomatic arch (ZL); d. right zygomatic arch (ZR)	A. Mean differences between the two superimposition techniques
B. Mean differences between repeated AC superimposition measurements
C. Mean differences between repeated LZ superimposition measurements	Mean distances measured between the models (mm): 1. 0.20 - 0.37 (SD: 0.08 - 0.16); 2. 0.20 - 0.45 (SD: 0.09 - 0.27)	A. Mean differences (mm): a. 0.12 ± 0.19; b. 0.19 ± 0.12; c. 0.15 ± 0.18; d. −0.17 ± 0.13
B. Mean differences (mm): a. 0.02 ± 0.09; b. 0.01 ± 0.07; c. −0.07 ± 0.12; d. 0.04 ± 0.09
C. Mean differences (mm): a. −0.07 ± 0.25; b. 0.04 ± 0.24; c. 0.14 ± 0.10; d. 0.04 ± 0.09	
Nguyen et al.,
Am J Orthod Dentofacial Orthop (2017)	1. Absolute mean surface distance of the registered models on plates and screws, calculated at 3 regions: a. Chin, b. Symphysis, c. Lower contour of the third molar crypt
2. Reproducibility of the combined chin+symphysis regions measured through ICC and mean absolute distances of the entire surface of T2 registered mandibular models by two operators	NA	1. Absolute mean surface distance (mm): a. 0.37 ± 0.16; b. 0.40 ± 0.14; c. 1.94 ± 0.06
2. ICC = 0.998 (95% CI [0.995–1.000])	NA	
Ruellas et al.,
Am J Orthod Dentofacial Orthop (2016a)	Differences between corresponding landmark distances from T0-T1 measured through the two superimpositions	A. Precision and B. reproducibility of each technique measured as differences in Euclidean distances of corresponding landmarks	Mean differences (mm): 0.35 - 0.39 (SD: 0.23 - 0.24)	A. Mean differences (mm): 0.36 - 0.42 (SD: 0.21 - 0.24)
B. Mean differences (mm): 0.31 - 0.44 (SD: 0.16 - 0.28)	
Ruellas et al.,
PLoS One (2016b)	Difference of corresponding landmark distances between T0-T1 calculated through superimposition on 3 different reference regions, compared to direct measurements of landmark movements from a point considered stable	NA	NA (Mean values provided were outside of the Limits of Agreement range)	NA	
Weissheimer et al.,
Int J Oral Maxillofac Surg (2015)	Visual inspection of the superimposition technique and trueness assessment through visualisation of 3D colour maps	Visual inspection of the effectiveness of the technique through superimposition of reoriented identical models	Highest distance between corresponding anterior cranial base references is less than 0.5 mm for growing and non-growing patients	Highest distance between identical, reoriented anterior cranial bases was less than 0.25 mm	

Table 5 Conclusions and limitations of the included studies.

Study	Conclusions	Limitations	
Almukhtar et al.,
PLoS One (2014)	No differences between Voxel-based registration and Surface-based registration.
High inconsistency between VBR and SBR regarding soft tissues.	I. No method error.
II. In SBR, hard and soft tissues were superimposed separately whereas in VBR, hard and soft tissues were all superimposed at once.	
Bazina et al.,
Am J Orthod Dentofacial Orthop (2018)	The Dolphin 3D software seems to work properly for voxel-based registration in the anterior cranial base.	I. The original change that occurred over time is not reported.
II. ICC values were calculated from only 10 patients and for the average of all measurements.
III. There was no assessment of the reproducibility of each individual measurement/case.
IV. The type of ICC used is not reported.	
Cevidanes et al.,
Dentomaxillofac Radiol (2005)	The technique shows acceptable reproducibility in the assessment of relatively unaltered structures.	I. There were relatively large interobserver errors compared to the detected changes.
II. The actual measured changes were originally small (<0.8 mm).	
Cevidanes et al.,
Am J Orthod Dentofacial Orthop (2009)	The technique provides reproducible 3D assessment of growing patients.	I. Small sample size that did not allow statistical comparisons.	
Gkantidis et al.,
PLoS One (2015	Superimposition of 3D surface models created from voxel data can provide accurate, precise and reproducible results when appropriate references are used.
Superimposition on BZ could be an alternative to AC.	I. CT data were used.
II. No assessment of individual measurements regarding reproducibility.	
Ghoneima et al.,
Orthod Craniofac Res (2017)	Surface-based and Voxel-based superimposition methods using the anterior cranial base as reference seem to be reproducible whereas Landmarks-based superimposition is less reproducible.	I. The original change that occurred over time is not reported.
II. ICC values were calculated from only 10 patients and for the average of all measurements. There was no assessment of the reproducibility of each individual measurement/case.
III. The type of ICC used is not reported.
IV. The time span between serial images is not reported.	
Häner et al.,
Orthod Craniofac Res (2019).	The Dolphin voxel-based superimposition technique exhibited adequate performance in growing patients, in terms of efficiency, cranial base matching, and reproducibility.
The segmentation error was also acceptable in most cases.	I. The trueness of the voxel-based superimposition was assessed through visual inspection of corresponding reference structures in 2D.
II. The original changes between T0 and T1 were relatively limited, though no relation was evident between the amount of change and the error of the process.	
Koerich et al.,
Int J Oral Maxillofac Surg (2016)	The technique shows high precision and reproducibility tough these were assessed in relatively unaltered structures. Furthermore, differences between reoriented dry skulls were larger than expected.	I. The changes of the structures that were evaluated were quite small (<0.3 mm).
II. Differences between the serial images of reoriented dry skulls were higher than those of actual serial scans.
III. Samples from different machines were tested.	
Koerich et al.,
Angle Orthod (2017)	The technique shows moderate reproducibility in the assessment of relatively unaltered structures.	I. Relatively large interobserver errors compared to the detected changes.
II. The changes measured were small (<0.9 mm).	
Lemieux et al.,
Am J Orthod Dentofacial Orthop (2014)	The landmark-derived maxillary plane cannot be assessed through the present methodology.	I. The main outcome is not suitable for the assessment of the superimposition result because it remains unaffected by the superimposition itself.
II. The landmark identification error is not thoroughly assessed for individual cases.	
Nada et al.,
PLoS One (2011)	This technique might show good trueness and reproducibility.
Registration on the left zygomatic arch seems to be less accurate, but it might still be clinically acceptable and reproducible.	I. Only structures considered stable were evaluated and thus the measured changes were small.
II. Only mean values are provided and analysed and thus possible larger individual differences are ignored.	
Nguyen et al.,
Am J Orthod Dentofacial Orthop (2017)	The chin and the symphysis region might be an anatomically stable reference area for mandibular superimpositions, whereas the third molar region displayed a higher instability.
The chin+symphysis area seems to provide reproducible results.	I. The bone plates and screws were confirmed to be immobile clinically, but their stability in space was not tested (e.g., through best fit registration).
II. The areas identified as stable were located at the same place where the superimposition reference area was.
III. Only average measures were used to assess all outcomes. There was no assessment of individual cases.
IV. Reproducibility outcomes were tested assessing the whole mandibular surface.
V. The performance of the chin+symphysis area was shown only for 1 subject.	
Ruellas et al.,
Am J Orthod Dentofacial Orthop (2016a)	No clear evidence is provided that the 2 regions of maxillary registration show similar results and adequate intraobserver and interobserver reproducibility values for growing patients.	I. The changes measured were originally small, except from landmarks 2 and 6 where the error was greater.
II. In individual cases the amount of differences was not small compared to the original changes.
III. No detailed information is provided (e.g., Bland Altman plot for every variable and results for any landmarks and every coordinates). Only means of different variables were assessed.
IV. No comparative statistics.	
Ruellas et al.,
PLoS One (2016b)	The body of the mandible might show better agreement with direct measurements from a point considered stable, compared to the modified Björk superimposition.	I. Results from one superimposition technique (Björk) are not reported.
II. The gold standard reference values were not reliable since one geometrical point that was speculated to be stable was used to generate them. However, landmark point identification error is expected to be high in this case and this was not evaluated.
III. Two cases were not included in the analysis.
IV. Reported mean values were outside of the Limits of Agreements.	
Weissheimer et al.,
Int J Oral Maxillofac Surg (2015)	The software seems to be user-friendly and might work properly for voxel-based registration in the anterior cranail base, both for growing and non-growing patients.	I. No results quantification.
II. No method error.
III. No descriptive and comparative statistics.
IV. Only data from 2 patients shown.	

Koerich et al. (2016) investigated the precision and the reproducibility of one superimposition method in the maxilla and one in the mandible. As superimposition references for the maxilla, they used two areas (zygomatic process; palate) and for the mandible three areas (symphysis; corpus; part ramus). The sample for this study included two dry skulls and 15 non-growing patients. Different machines and acquisition parameters were used in the dry skulls and the actual patients. This technique has shown excellent precision and reproducibility, although the evaluated regions are considered relatively unaltered. Surprisingly, the distances obtained from the superimposition of the two dry skulls were higher than expected and than those acquired from the superimposition of actual serial scans. This study had high risk of bias and high applicability concerns.

Koerich et al. (2017) also assessed the precision and reproducibility of a 3D mandibular voxel-based superimposition in 24 growing patients. To test the performance of this technique, distances between serial models at five mandibular regions located at the outer surface of the mandible were measured. Although the assessed structures were originally relatively unaltered the inter-observer variation was larger than expected. Thus, this mandibular technique showed moderate precision and reproducibility in the assessment of relatively unaltered structures. The study had high risk of bias and high applicability concerns.

Nada et al. (2011) evaluated the accuracy and reproducibility of a voxel-based registration of CBCT models on two different regions: the anterior cranial base and the zygomatic arches. Data were collected from 16 non-growing patients. Changes were measured afterwards on four anatomical regions, which were deemed stable: the anterior cranial base, the forehead, the left zygomatic arch, and the right zygomatic arch. The accuracy and reproducibility of this technique seems to be high, although the original changes measured were small. The superimposition on the left zygomatic arch appears to be a valid alternative to that on the anterior cranial base in non-growing patients. The added advantage is that it can be used in images with smaller field of view, and thus, lower radiation. However, individual changes were not reported and only mean values were assessed. The study had high risk of bias and applicability concerns.

Nguyen et al. (2018) searched for stable anatomical regions in the mandible by superimposing CBCTs of growing patients on bony plates and miniscrews. They concluded that the chin and symphysis region might be anatomically stable, whereas the third molar region displayed higher instability. However, among other limitations, the bone plates and screws were confirmed to be immobile clinically, but their stability in space was not tested. The study had high risk of bias and high applicability concerns.

Ruellas et al. (2016a) aimed to identify stable maxillary superimposition references for growing patients. The precision and reproducibility of two different maxillary regions were tested on a sample of 16 patients. To quantify changes, distances between corresponding landmarks at pre- and post-treatment models after registration were assessed. However, the absence of comparative statistics and the evaluation of average effects of different variables did not allow for a clear conclusion. The study had high risk of bias and high applicability concerns.

Ruellas et al. (2016b) evaluated three reference regions for mandibular superimposition using a sample of 16 growing patients. Following superimposition of the serial scans and analysis of the distances between corresponding landmarks, the body of the mandible seemed to show better agreement with direct measurements from a point considered stable, when compared to the modified Björk technique. The performance of the Björk technique was not reported in the study, due to software performance issues. The reporting of the results was poor, since the presented mean values were outside of the provided limits of agreement. The study had high risk of bias and high applicability concerns.

Weissheimer et al. (2015) also performed voxel-based superimposition on the anterior cranial base in serial 3D CBCT models of growing and non-growing patients. The assessment of the accuracy was done through visual inspection of the congruence of the anterior cranial base between serial models using colour coded distance maps. Through this, it was established that the highest distance was less than 0.5 mm. Thus, it seemed that the software works properly and the anterior cranial base is a stable superimposition reference in both growing and non-growing patients. However, the study has no descriptive or comparative statistics. Merely data from two patients were shown. Thus, the study has shown high risk of bias and applicability concerns.

Landmark-based registration

Lemieux et al. (2014) evaluated the trueness of a maxillary superimposition plane using the nasomaxillary complex as reference. CBCTs of 30 growing patients were superimposed on a maxillary superimposition plane formed by the nasion, the bilateral infraorbital foramina and the incisive foramen. However, the performance of this landmark-derived maxillary plane cannot be assessed through the present methodology. The study is graded with high risk of bias and high applicability concerns.

Comparison of different registration techniques

Since now, a single study compared the accuracy of voxel-based registration and surface-based registration for the 3D assessment of surgical change following orthognathic surgery (Almukhtar et al., 2014). The sample included only non-growing patients. The surface-based registration on hard tissues was performed on the anterior cranial base; as for the registration on soft tissues, the forehead and the eyes were selected. Regarding the voxel-based registration, the structures described above were chosen, but in this case hard and soft tissues were used simultaneously as superimposition references. The assessment of accuracy in this study was tested via measurements on the anterior cranial base for the hard tissues and on the forehead for the soft tissues. The mean absolute distances of surface models in hard tissues did not differed much between the voxel and the surface-based registrations, but this was not the case for the soft tissues. This can be attributed to the differences in the superimposition references used each time. The study showed high risk of bias and applicability concerns.

Gkantidis et al. (2015) investigated the accuracy, precision, and reproducibility of four surface-based and one landmark-based 3D superimposition technique. Pre-existing CT data from eight non-growing patients were analysed by three operators. To confirm the accuracy of each technique, the congruence of serial models was measured in three areas that were considered stable. For precision testing, the distances between four corresponding landmarks were quantified. The whole procedure was repeated to test reproducibility. The superimposition on the anterior cranial base showed acceptable outcomes that were comparable with the superimposition on both zygomatic arches. The study concluded that the superimposition of 3D surface models created from voxel data can provide accurate, precise, and reproducible results when appropriate references are used. Since this study used CT data, a similar study on CBCT data of non-growing patients would be required to confirm these findings. Therefore, this study had low risk of bias, but high applicability concerns.

Ghoneima et al. (2017) evaluated the reproducibility of landmark-based, surface-based, and voxel-based superimpositions, as well as their performance in matching duplicated scans. They superimposed CBCTs of 20 growing patients. The superimposition area for the landmark-based method was defined on seven homologous points on the frontal and zygomatic bones, for the surface-based method on the anterior cranial base, as well as for the voxel-based method. Regarding the results, the surface-based and voxel-based superimpositions seemed to be reproducible, whereas the landmark-based superimposition was less reproducible. Based on certain limitations the study was graded to have high risk of bias and applicability concerns.

Discussion

Due to the inherent limitations of 2D superimposition methods various scientific fields have turned their focus to the more thorough and accurate 3D imaging techniques, and worked to create more reliable, faster, and easy to handle software facilitating this purpose. This allowed researchers and clinicians to work with real size and shape 3D representations of anatomical structures. However, till today there is no single method that has been proved to be accurate, easy to use, and is widely accepted for superimposing 3D craniofacial radiographic images. This review performed a thorough, critical assessment of the recent literature and analysed 15 identified studies that tested one or more of the three available superimposition techniques for this; namely, the voxel-based, the landmark-based, and the surface-based technique. Overall, the study detected high heterogeneity and moderate study quality, emphasizing the urgent need for further relevant research in this rapidly expanding field.

The single previous systematic evaluation of the literature included six studies that were all published prior to 2017 and regarded only the anterior cranial base (Ponce-Garcia et al., 2018). In our review, we performed a more thorough selection process including all relevant studies for the whole craniofacial area and we managed to include 15 studies, though still the vast majority of these focuses on the anterior cranial base area.

For clarity reasons we divided the included studies in the following three major categories, based on the type of superimposition tested: landmark-based registration, voxel-based registration, and comparison of different registration techniques, which includes the surface-based registration. Landmark-based superimposition is relatively simple to use and understand, but small errors in the identification of landmarks may have a large negative impact on the results. This is especially true if a limited number of landmarks is used, but only then the method is simple and easy (Gkantidis et al., 2015; Becker et al., 2018). Only one study investigated exclusively a landmark-based superimposition technique (Lemieux et al., 2014). This was graded as high risk of bias and applicability concerns. Two further studies (Gkantidis et al., 2015; Ghoneima et al., 2017) that compared the landmark-based superimposition to other superimposition techniques (voxel- or surface-based) concluded that the landmark-based superimposition was inferior to the others. Overall, there is a lack of well-designed studies to support the use of landmark-based superimposition. The existing weak evidence indicates that this technique might be unreliable, especially when few landmarks are used as superimposition reference. Thus, the use of landmark-based superimposition remains questionable.

Most of the included studies (11/15) investigated a voxel-based superimposition technique (Cevidanes et al., 2005; Cevidanes et al., 2009; Nada et al., 2011; Weissheimer et al., 2015; Ruellas et al., 2016a; Ruellas et al., 2016b; Koerich et al., 2016; Koerich et al., 2017; Bazina et al., 2018; Nguyen et al., 2018; Häner et al., 2020). This type of superimposition utilizes the original volume generated from a 3D radiographic scan and no further data processing is required prior to the superimposition. That might also be a reason why most studies focused on this type of superimposition. Most of the studies that investigated a voxel-based superimposition technique (n = 6) used cranial base structures as superimposition reference, whereas two studies used maxillary and four mandibular sites. Thus, the cranial base is the most widely tested and supported reference for voxel-based superimposition, but until now the quality of evidence for this ranges from low to moderate. More work needs also to be performed to find alternative reference areas that might be applicable is smaller field of view scans, reducing the required radiation amount. So far, two studies have investigated this issue (Nada et al., 2011; Gkantidis et al., 2015), but they both had high applicability concerns. Regarding the maxillary and the mandibular areas the amount of existing evidence is lower and of low quality. Overall, nine of the included studies in this category had high risk of bias and high applicability concerns and two unclear. There is no study graded with low risk of bias or low applicability concerns. Hence, there is an urgent need for well-designed studies with low risk of bias and low applicability concerns to support the voxel-based superimposition techniques.

There was no study that focused only on surface-based superimposition. Surface-based registration compares the triangular representations of corresponding 3D surface geometries on the models. This technique might show adequate accuracy, it is less sensitive and time-consuming, and has increased post-processing capabilities (Gkantidis et al., 2015). Three studies that compared different registration techniques (Almukhtar et al., 2014; Gkantidis et al., 2015; Ghoneima et al., 2017) included a surface-based technique. Two of them had high risk of bias and all of them had high applicability concerns. The study of Gkantidis et al. (2015) showed low risk of bias, but high applicability concerns, and did not support the use of landmark-based superimposition but showed acceptable results for surface-based superimposition. Ghoneima et al. (2017) did not recommend the use of landmark-based superimposition as well, but they showed promising results for voxel-based and surface-based superimposition. Almukhtar et al. (2014) provided similar and promising results for voxel-based and surface-based superimposition of hard-tissues. Thus, the three above studies support the surface-based superimposition on the anterior cranial base structures. Two of them also support the voxel-based superimposition (Almukhtar et al., 2014; Ghoneima et al., 2017), whereas other two do not support the landmark-based superimposition (Gkantidis et al., 2015; Ghoneima et al., 2017). However, the quality of evidence for the above outcomes ranges from moderate to low.

Overall the literature supports the use of voxel-based and surface-based superimposition techniques, though the existing evidence is not yet strong. Because of the limited amount of well-designed studies, further research is needed to confirm the present findings. It seems that these techniques show better accuracy and are less operator-sensitive compared to the landmark-based superimposition. A limitation of the surface-based registration is the lack of information concerning inner structures as only the surface information is available after processing. Furthermore, an additional step is required to segment the surface model of interest from the original 3D volume and this might induce error (Häner et al., 2020). The voxel-based registration is applied to the original volumetric data derived from a 3D radiographic scan, and thus, this might be advantageous in terms of less error prone steps required to achieve model registration. However, after the registration of serial volumes, surface models are usually required for thorough assessment and visualization of the results. Thus, this possible source of error is not fully eliminated also through this method. Furthermore, the surface models are widely used in various other scientific disciplines and in the industry, leading to well-developed methods and software applications for data processing and evaluation. Thus, the acquisition of accurate surface models from the original volume is quite important to take advantage of these possibilities for data processing and visualisation and will also facilitate accurate surface model superimposition techniques (Henninger et al., 2019).

Though a significant amount of studies was identified, a limitation of the present study is that the heterogeneity of the included studies is high, and the quality of the available evidence is limited. This can be attributed to the fact that the field has been developed in the last few years and gained much attention only recently.

Conclusion

The fast evolution of 3D superimposition techniques has provided a key element in the toolkit of relevant fields to evaluate craniofacial changes following growth or treatment. Due the high heterogeneity and the moderate to low quality of the included studies, few valid conclusions can be drawn. Most of the available studies had methodological shortcomings and high applicability concerns. Therefore, no clear recommendation could be given at present for proper methods used for 3D-superimposition of craniofacial skeletal structures. At the moment, certain voxel-based and surface-based superimpositions seem to work properly and to be superior compared to landmark-based superimposition. However, further research is necessary to develop and properly validate these techniques on different samples, through studies of high quality and low applicability concerns.

Supplemental Information

Supplemental Information 1 Rationale for the study

Click here for additional data file.

Supplemental Information 2 Detailed description of the search performed in various databases

Click here for additional data file.

Supplemental Information 3 PRISMA checklist

Click here for additional data file.

Additional Information and Declarations

Competing Interests

Author Contributions

Data Availability

The authors declare there are no competing interests.

Daniel Dinh-Phuc Mai, Sven Stucki and Nikolaos Gkantidis conceived and designed the experiments, performed the experiments, analyzed the data, prepared figures and/or tables, authored or reviewed drafts of the paper, and approved the final draft.

The following information was supplied regarding data availability:

All the data used in this systematic review are available in Tables 1–5. The specific search strategies used to locate the literature are available in the Appendix S1.

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
