# Peer review of "Assessment of methods used for 3-dimensional superimposition of craniofacial skeletal structures: a systematic review"

_PeerJ, doi:10.7717/peerj.9263_

## Round 0.1 · original submission · Major Revisions

Major Revisions

Dear authors,

Thank you for your patience during the review process. I have now received reviewers' comments and have appended them within.
Please also make sure that your manuscript follows the PRISMA guidelines on par with the extension for diagnostic test accuracy review. Details can be found at http://www.prisma-statement.org/Extensions/DTA [JAMA. 2018;319(4):388-396], DOI: 10.1001/jama.2017.19163.

I hope that you are willing to take all actions needed to revise the manuscript accordingly.

With kind regards,
Despina Koletsi
Academic Editor, PeerJ

Reviewer 1 ·

Basic reporting

• The article is well-written; all sections of the article are clearly written, and the message to the reader is direct. English language is professional and fluent.
• The “Title” is short, accurate and provides the reader with a direct message of what to expect.
• The literature is well referenced and relevant.
• Figures and Tables are relevant, well-designed, of high quality, well labelled and described.
• The supplemental Materials (“Appendix_1”, “PRISMA_Checklist” and “Rational_for_the_study” file) are informative and support the validity of the article.
• Please spell out numbers at the start of sentences [e.g. line 176: Twenty-four instead of 24, line 190, line 200, etc.]. It would be better to add an introductory word or phrase, or even rephrase the sentence, so that you can avoid spelling out a large number [e.g. line 174: “2’540 studies…].
• Maybe the term “Bibliography” could be replaced by the term “References” concerning the corresponding section.

Experimental design

• The research question is well-defined. The “inclusion/exclusion criteria” are well-established. All key components prior to starting the review have been decided and rigorously cited (i.e. studies included, study sample, types of participants, type of intervention, primary outcome, comparator/control group etc.). Methods were described with sufficient detail and information to replicate.

Validity of the findings

• All underlying data have been provided, they are informative and support the validity of the results.
• The development and run of the electronic searches are meticulously described in the “Appendix_1” file. A comprehensive list of the key terms, related to each key component is present, ensuring that all relevant trials were identified.
• The selected databases were screened by two authors, and discussed between all authors until a consensus was reached, thus establishing inter-rater reliability.
• Thorough analysis and interpretation of the results was conducted.
• The conclusions are well-stated, they are linked to the original research question and summarize the outcomes of the study.

·

Basic reporting

- The authors have submitted an overall well written and well conducted systematic review.
- The literature review is sufficient and supports performing this review.
- Lines 58-60: Text needs to be revised here. The authors briefly mention the advantages of 3D over 2D but are emphasizing head positioning, which is less relevant to the topic of superimposition. The text here does not offer much to the manuscript.
- Lines 60-62: This text lacks citations to support what is stated. I am particularly referring to the sentence “Moreover, the handling and processing of these 3D data is more complex and expensive.” Please revise. My impression is that the authors are attempting to provide some background information on 3D imaging compared the 2D imaging. This part (lines 57-62) needs to be revised.
- Lines 65-66: Larger field of view means more radiation because anatomic structures that absorb higher amounts of radiation are included in the field of view. It would be better if this is mentioned and supported by a reference.
- Lines 66-68: The new methods that the authors are mentioning do not require a large field of few. This is indirectly related to the amount of radiation. The sentence implies that the new methods are directly related to less radiation.
- Line 74: What do the authors mean by “more comfortable” treatments?
- The authors have identified a gap in the literature and, thus, the rationale for conducting this systematic review is justified.
- Lines 147-149: Precision and sensitivity of a systematic search of the literature are by definition opposite, negatively correlated terms. Please revise wording.

Experimental design

- The research question is well defined and the search strategy is provided in detail as supplement.
- Line 97: Please replace “Pubmed” with “Medline (via Pubmed).
- The authors selected the databases recommended by the Cochrane Collaboration for systematic reviews of RCTs, in addition to Google Scholar, Grey literature and Unpublished literature. It has been suggested that the recall percentage of articles may increase by 3% if Web of Science is added to a systematic search, increasing however the NNR (number needed to read) (see: Wichor M. Bramer, Melissa L. Rethlefsen, Jos Kleijnen, Oscar H. Franco. Optimal database combinations for literature searches in systematic reviews: a prospective exploratory study. Syst Rev. 2017; 6: 245. Published online 2017 Dec 6. doi: 10.1186/s13643-017-0644-y) Although their search strategy is very good, the authors may want to briefly discuss their selection of databases.

Validity of the findings

- The study is novel and provides very useful insights in the field of 3D superimpositions, which are becoming increasingly relevant in clinical practice.
- The study results are thoroughly discussed and valuable implications for future research are made.

·

Basic reporting

The introduction is nicely written. However does not really provide other insights compared to the systematic review done by Ponce-Garcia et al 2018.
line 52 the structural method. This is the golden standard for 2 D. So why do you mention it here? How could this be translated to 3D. What should be the golden standard for 3D? How does that compare to 2D?
I also miss a paragraph on what the problems are between non growing and growing persons.
You write that since in 3D one would not want to include for example sella because of the higher radiation dose. However you do not come back on this in your review. Did researchers find a way around it or is that why most research concentrates on the anterior cranial base.....
What would applicability include. This is something that returns often later in your manuscript. However you do not explain this.

Experimental design

The purpose of the review is really wide and it is actually logical that a previous systematic review researched mostly the anterior cranial base. See also your reference to the structural method by Björk. other areas maybe maxilla or mandible. In case you would want to do that specify. Furthermore in the systematic review you are expanding on growth versus non growth. However this was not covered in your aim.
I miss that you used the prisma reporting guidelines. Did you use them?
line 103: You are looking for evidence in a systematic review. Why would you want to include case reports opinions and other such things?
Line 105: why did you choose a study sample of 3 or more? Of course that is not an amount that you could do statistics on?
line 108: why did you choose serial craniofacial CT/CBCT? you do not write growth/non growth in your aim?
line 113 Actually in the 3 techniques you name it does mathematically not matter how you superimpose when you compare 3D distances or angles. The only way to really mathatically test whether a superimposition works is through distance kits and volume differences. So The primary outcome alone will provide a mixed bag of studeis.
line 117 I actually do not understand the comparator control group. Please make full sentences of what you mean here.
line 119 how do you now that the unit of analysis was the measured distance /angle? this excludes distance kits/tables or volumes.
line 121: this is also strange. You are also describing in your results section growth. For example Studies done in patients with craniofacial malformations should be excluded because they have a different growth pattern.
142 heterogeneity as I understand this as done in and with the Quadas-2 tool. Please explain
146 bias as I understand this as done in and with the Quadas-2 tool. Please explain
line 162. You used the Quadas-2 tool. It requires that you make it specific to this systematic review. please explain that.
Line 166 you write that it uses happy and sad smiles. What is in between? Does no moderate exist. What is the scale of that? Please also include that in the table (the legenda that is)

Validity of the findings

line 174 has a high ' ?
line 177 what does irrelevant mean? please explain
In the quality assessment you start now with a distinction between risk of bias and applicability concerns. Where did the heterogeneity go?
line 190 so what do you define as applicability concerns? please define
I miss any numbers in the quantitative synthesis. A statistically significant outcome maybe sub millimiter and the other way round.
I also do not understand what information I should get from these small summeries.
214 How can it be that you start out with 3 superimposition techniques while only 2 were left here (and in the Ponce-Garcia et al 2018 they did include one)
line 316 you do compare the different registration techniques but you present only 2 before hand and are reporting o 3 that is quite confusing.
Discussion.
you refer to providing guidelines in the aim but that was probably not possible. so omit it from your aim.
line 363. Well this is logical to select the overall picture. You want to include maxilla and mandible in the systematic review as well but I do not understand why you think that is comparable to the anterior cranial base. It depends on what part you want to focus on.
Again what are applicability concerns
Figure one. How can it be that you excluded records. You have no exclusion criteria according to your material and methods.
Table I. I do not find the explanation of reference standards or index tests. It is seen in risk og bias and applicabiliay concerns. What does it refer to? What does flow and timing mean/include?

Additional comments

I recommend that you focus your review on growing subjects or non growing subjects. When you use growing patients please distinguish between growth and result of treatment or a mixed bag.

---

## Round 0.2 · accepted · Accept

The authors have now addressed reviewers' comments adequately and no further revision cycle is necessary. The manuscript may be accepted in its current form.
Thank you